# HUMAN-IN-THE-LOOP TEST-TIME DOMAIN ADAPTATION FOR OBJECT DETECTION

## ABSTRACT

Prior to deployment, an object detector is trained on a dataset compiled from a previous data collection campaign. However, the environment in which the object detector is deployed will invariably evolve, particularly in outdoor settings where changes in lighting, weather and seasons will significantly affect the appearance of the scene and target objects. It is almost impossible for all potential scenarios that the object detector may come across to be present in a finite training dataset. This necessitates continuous updates to the object detector to maintain satisfactory performance. Test-time domain adaptation techniques enable machine learning models to self-adapt based on the distributions of the testing data. However, existing methods mainly focus on fully automated adaptation, which make sense for applications such as self-driving cars. Despite the prevalence of full automated approaches, in some applications such as surveillance, there is usually a human operator overseeing the system's operation. We propose to involve the operator in domain adaptation to raise the performance of object detection beyond what is achievable by fully automated adaptation. To reduce manual effort, the proposed method only requires the operator to provide weak labels, which are then used to guide the adaptation process. Furthermore, the proposed method can be performed online, facilitating its applications in scenarios where inference and domain adaptation must be carried out simultaneously. Our experiments show that the proposed method outperforms existing works, demonstrating a great benefit of human-in-the-loop test-time domain adaptation.

## 1 INTRODUCTION

Object detection is a task that involves precisely localising and categorising objects within an image. It has many applications in autonomous driving (Han et al., 2021), surveillance (Lu et al., 2023), and augmented reality (Li et al., 2020). The deployment of an object detector typically includes three main steps. Firstly, a large-scale dataset must be collected and annotated, providing the bounding boxes and object categories for the objects of interest. Next, this annotated dataset is used to train an object detector. Finally, the object detector is deployed into a desired system to effectively perform real-time object detection.

However, while the training dataset is important for preparing an object detector, it may not cover all possible scenarios that the detector may encounter during its operation. This incomplete coverage is attributed to the various environmental conditions that can arise, such as different times of day, weather, and seasons. These factors cause the image appearance to differ from the training dataset, leading to a significant decline in detection accuracy. A solution to the problem is to continuously capture new data and adapt the system (Doan et al., 2020; Mirza et al., 2022; Wang et al., 2022). However, incorporating new data presents a substantial challenge due to the absence of labels within this new data, making the adaptation of the object detector a challenging task.

A potential solution for this issue is unsupervised domain adaptation (UDA) (Chen et al., 2018; RoyChowdhury et al., 2019; Li et al., 2022), which formulates the training dataset as the source domain and the newly acquired data as the target domain. The goal of UDA is to minimise the domain discrepancy in the feature space. Popular UDA methods include adversarial learning (Chen et al., 2018; 2021; Pasqualino et al., 2021), optimal transport (Lee et al., 2019; Xu et al., 2019), and pseudo-labelling (RoyChowdhury et al., 2019; Li et al., 2022; Mattolin et al., 2023). However, a

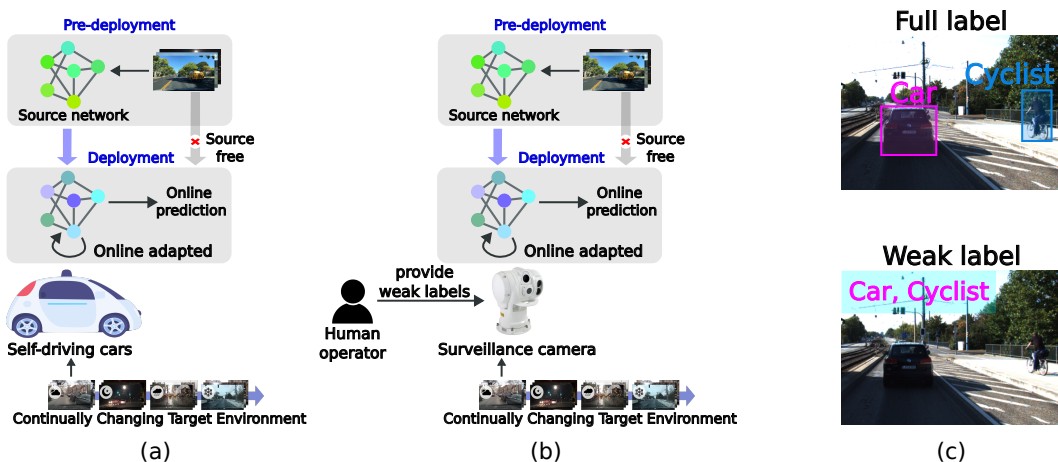

Figure 1: **(a)** Previous works have focused on developing fully autonomous solutions, primarily for self-driving vehicles. **(b)** Our approach, however, is proposed for visual surveillance, which are typically monitored by an operator. Therefore, our method will take advantage of the operator's involvement in the adaptation process. **(c)** The definitions of full and weak labels: A full label includes bounding boxes and object categories. A weak label only indicates which object categories are present in the image. By only requiring weak labels, our method reduces the amount of labour needed significantly.

| Approach | Source data | Target data | Online adaptation | Human in the loop |
|---|---|---|---|---|
| Unsupervised domain adaptation (Chen et al., 2018) | ✓ | ✓ | ✗ | ✗ |
| Weakly supervised domain adaptation (Inoue et al., 2018) | ✓ | ✓ | ✗ | ✓ |
| Test-time domain adaptation (Mirza et al., 2022) | ✗ | ✓ | ✓ | ✗ |
| Human-in-the-loop test-time domain adaptation | ✗ | ✓ | ✓ | ✓ |

Table 1: A comparison of human-in-the-loop test-time domain adaptation with related approaches.

limitation of UDA is the offline setting: target data has to be acquired first before adapting the model for multiple epochs, whereas many practical applications necessitate domain adaptation to be done online. In addition, UDA requires complete access to the source domain, raising serious privacy and security concerns. Recent reverse engineering techniques have demonstrated that it is possible to use a limited amount of information about the data to fully recover the original data (Mahendran & Vedaldi, 2015; Dosovitskiy & Brox, 2016; Pittaluga et al., 2019). In data-driven approaches, data can be viewed as a vital asset of businesses; thus storing source data in deployed systems is indeed a hazardous undertaking.

To address the issues of UDA, test-time domain adapation (TTA) attempts to adapt the object detector to the target domain without the need for the source dataset (Wang et al., 2021; 2022; Mirza et al., 2022). Recent studies have demonstrated that TTA can be highly effective in image classification by adapting the model with pseudo-labelling and entropy minimisation (Chen et al., 2022; Wang et al., 2021). However, TTA requires full access to the target data while in practice, the target data is usually in the form of stream, resulting in the target distribution continually evolving. To address this challenge, continual TTA (CoTTA) (Wang et al., 2022) and Dynamic Unsupervised Adaptation (DUA) (Mirza et al., 2022) have been proposed. These methods only require an incoming target sample to adapt the model, making them suitable for online adaptation. The effectiveness of CoTTA has been demonstrated in image classification and image segmentation, while DUA has been proven to be effective in object detection.

Despite their great potential, CoTTA and DUA are striving for a fully autonomous solution, which is suitable for applications such as self-driving cars. However, there are some applications, such as surveillance, which usually have a human operator overseeing the system (Bloisi et al., 2016). This raises a question of whether we should involve this operator to TTA. One benefit of human-

in-the-loop TTA is to revise the pseudo-labels used for adapting the object detector. As shown in previous works (Li et al., 2022; Chen et al., 2022; Litrico et al., 2023), pseudo-labelling is an effective approach for domain adaptation. However, if pseudo-labels are noisy, the object detector's error will accumulate, leading to a decline in the detection accuracy. Therefore, humans can be another reliable annotator for revising pseudo-labels. Efficient use of human contributions with minimal demands on labour cost in TTA is thus a critical objective. Our idea is illustrated in Fig. 1

**Contributions** This paper proposes the inclusion of humans in the adaptation of object detectors. Our method, dubbed human-in-the-loop test-time domain adaptation (HL-TTA), uses weak labels provided by humans to guide the domain adaptation during the testing phase. As HL-TTA only requires weak labels to be effective; its demand on labour cost is therefore minimal. Furthermore, HL-TTA can be done online, where the target test data is in the form of stream, allowing inference and domain adaptation to be performed simultaneously. The experiments show that with only a few target test images, the HL-TTA outperforms existing fully autonomous solutions. We hope that this encouraging result will motivate further research in HL-TTA.

## 2 RELATED WORK

This section will review three main approaches to domain adaptation for object detection: unsupervised domain adaptation (UDA), weakly supervised domain adaptation (WSDA), and test-time domain adaptation (TTA). Also, we will discuss the novelty of HL-TTA in comparison to these approaches, which are succinctly outlined in Table 1.

### 2.1 UNSUPERVISED DOMAIN ADAPTATION

Given a labelled source dataset and an unlabelled target dataset, UDA seeks to adapt an object detector to perform accurately in the target domain. There are three main techniques in UDA for object detection: adversarial learning (Chen et al., 2018; 2021; Pasqualino et al., 2021), optimal transport (Lee et al., 2019; Xu et al., 2019), and pseudo-labelling (RoyChowdhury et al., 2019; Li et al., 2022; Mattolin et al., 2023).

Adversarial learning attempts to minimise the domain discrepancy in the feature space. To this end, domain adaptive Faster-RCNN (Chen et al., 2018) employs gradient reversal layers (Ganin & Lempitsky, 2015) in their adversarial learning framework to align the feature and instance distributions of source and target domains. This concept is further improved in (Chen et al., 2021), which aligns the source and target distributions across different image scales. Additionally, adversarial learning has been explored in anchor-free object detection techniques (Pasqualino et al., 2021; 2022). In comparison to adversarial learning which solves a minimax optimisation problem, optimal transport instead minimises Wasserstein distance between source and target domains. For instance, (Lee et al., 2019) employs the sliced Wasserstein distance to address high-dimensional issue in the feature space and (Xu et al., 2019) considers the duality of the Wasserstein distance, which can be approximated by neural networks under certain conditions. Another approach that has been shown to be effective is pseudo-labelling. To generate reliable pseudo-labels for target dataset, (RoyChowdhury et al., 2019) fuses the results of detection and tracking as well as proposes a label smoothing technique. To further improve the detection performance, some recent methods integrate pseudo-labelling to other strategies. For instance, (Li et al., 2022) incorporates pseudo-labelling to student-teacher architecture and (Mattolin et al., 2023) combines pseudo-labelling with domain mixing techniques.

Despite its great potential, UDA is done offline, which is not suitable for many practical applications that require online domain adaptation. Furthermore, the need for access to the source data can be a shortcoming in terms of privacy and security (see Sec. 1). To address this, our HL-TTA can be performed online and is a source-free method, thus avoiding any privacy and security risks.

### 2.2 TEST-TIME DOMAIN ADAPTATION

TTA attempts to adapt the model in an online manner without using the source data. Some interesting TTA works include Tent (Wang et al., 2021) which proposes to update batch normalisation layers using entropy minimisation, CoTTA (Wang et al., 2022) which uses teacher-student architecture with pseudo-labelling to adapt the model, and DDA (Gao et al., 2023) which employs a diffusion process

to transform the appearance of images from the target domain to resemble the source domain. Since these methods are only tested in image classification, DUA (Mirza et al., 2022) shows that TTA can be effectively applied to object detection by introducing a momentum decay parameter to stabilise the domain adaptation process.

As alluded, existing TTA works aim for developing fully autonomous domain adaptation techniques, which are useful to applications like self-driving cars. However, there are some other applications, such as surveillance, which usually have an operator overseeing the systems (Bloisi et al., 2016). Therefore, our HL-TTA proposes to leverage this operator to generate more reliable pseudo-labels, which can be useful for TTA.

### 2.3 WEAKLY SUPERVISED DOMAIN ADAPTATION

The concept of human-in-the-loop domain adaptation for object detection has been explored in the literature, which is usually referred to as weakly supervised domain adaptation (WSDA). This approach involves asking annotators to provide weak labels for the target dataset (see Fig. 1c). Then, domain adaptation can be done using the source dataset with full labels and the target dataset with weak labels. Inoue et al. (2018) propose combining predictions of the source pre-trained detector with weak labels to generate high-quality pseudo-labels for target images. Xu et al. (2022) attempt to minimise the domain gap by using domain and weak label classifiers.

Through using weak labels, WSDA is shown to outperform UDA. However, WSDA also suffers drawbacks similar to those of UDA, i.e., the domain adaptation is done offline and the need to access the source data raises privacy and security concerns (see Sec. 1). Therefore, HL-TTA is proposed to overcome these challenges.

### 3 METHOD

This section will elaborate our methodology. To begin with, Sec. 3.1 outlines the formal problem definition. Subsequently, the HL-TTA framework is presented in Sec. 3.2, where we will explain the loss function for domain adaptation. Given the loss function, we will discuss how the domain adaptation can be achieved via updating batch normalisation (BN) layers in Sec. 3.3.

### 3.1 PROBLEM FORMULATION

Let $f(\cdot\,;\theta_0)$ with parameters $\theta_0$ be an object detector that has been trained on the labelled source dataset $(\mathcal{X}^{\mathbf{S}}, \mathcal{Y}^{\mathbf{S}})$, where $\mathcal{X}$ and $\mathcal{Y}$ are the sample space and label space. During its operation, the detector will carry out the online inference and adaptation on the unlabelled target data $\mathcal{X}^{\mathbf{T}}$. Specifically, at time step $t$, the target data $x_t \in \mathcal{X}^{\mathbf{T}}$ is given as an input to the detector $f(\cdot\,;\theta_t)$. Then, the detector $f(\cdot\,;\theta_t)$ must make an inference $\hat{y}_t = f(x_t; \theta_t)$ and adapt itself $\theta_t \rightarrow \theta_{t+1}$ for the next input $x_{t+1}$. The detector's performance is evaluated based on the predictions $\hat{y}_t$ from the online inference. It is important to emphasise that during each time step $t$, the adaptation only relies on the target data $x_t$. This choice aligns with the recommendation of (Yang et al., 2022) that $x_t$ should be deleted immediately after the adaptation to safeguard the privacy.

The impetus for online adaptation is derived from practical scenarios in which perception systems are constantly operating in ever-evolving environments, with input coming in the form of streaming data. Consequently, inference and domain adaptation must be done online. Previous studies (Mirza et al., 2022; Wang et al., 2022; Gao et al., 2023) have mainly focused on developing fully autonomous TTA solutions for applications such as self-driving cars or autonomous robots. However, in certain applications, such as surveillance, a human operator is usually needed to supervise the system (Bloisi et al., 2016). Therefore, our idea is to involve the operator in TTA. Specifically, let $\mathcal{C}$ be a set of $L$ object categories. For each target image $x_t$, the operator will provide a weak label $z_t = \{c_j\}_{j=1}^{M}$, where $c_j \in \mathcal{C}$ is the object category present in the image and $M$ denotes the total number of object categories in the image; see Fig. 1c. This weak label $z_t$ will then be leveraged to adapt the object detector's parameters $\theta_t \rightarrow \theta_{t+1}$ for the next input. As discussed in previous works (Cao et al., 2021; Vo et al., 2022; Zhong et al., 2020), providing weak labels instead of full labels will significantly reduce the amount of labour required.

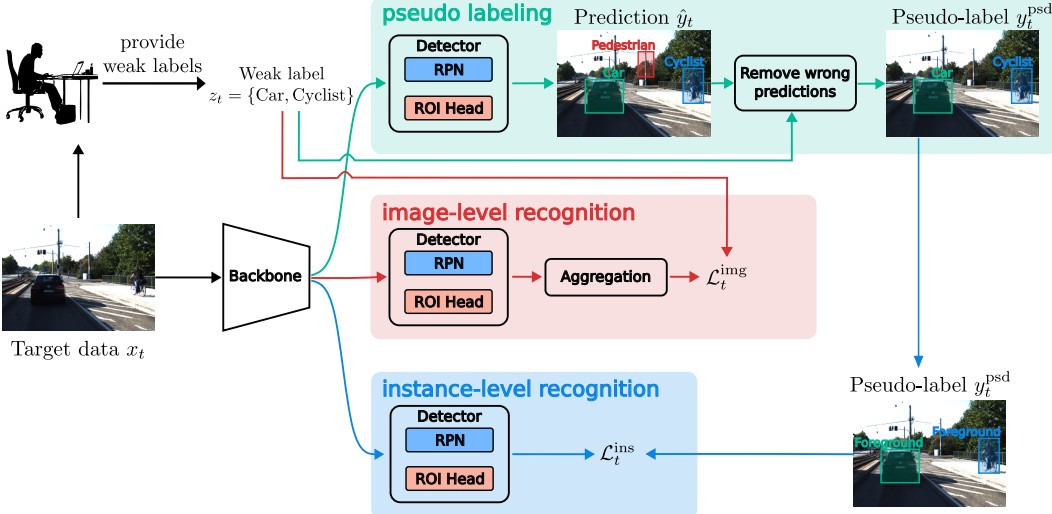

Figure 2: For an incoming target testing sample $x_t$, HL-TTA initially produces a prediction $\hat{y}_t$ and the operator is required to provide a weak label $z_t$ for it. Subsequently, using the prediction $\hat{y}_t$ and weak label $z_t$, a pseudo-label $y_t^{\text{psd}}$ is generated. Finally, the weak label $z_t$ and pseudo-label $y_t^{\text{psd}}$ are used as groundtruth for image-level recognition and instance-level recognition respectively.

## 3.2 FRAMEWORK

The overview of HL-TTA is shown in Fig. 2. Specifically, for the model $f(\cdot\,; \theta_t)$, HL-TTA adopts a two-stage object detection architecture Faster-RCNN (Ren et al., 2015) that includes a backbone, a region proposal network (RPN), and a detection head (ROI Head). Our HL-TTA consists of three main components: pseudo-labelling, image-level recognition, and instance-level recognition.

Using the prediction $\hat{y}_t$ and weak label $z_t$, pseudo-labelling will generate a pseudo-label $y_t^{\text{pds}}$. The pseudo label and weak label will be used to construct loss functions $\mathcal{L}_t^{\text{ins}}$ in instance-level recognition and $\mathcal{L}_t^{\text{img}}$ in image-level recognition. The final loss for domain adaptation will be

$$\mathcal{L}_t = \mathcal{L}_t^{\text{ins}} + \alpha.\mathcal{L}_t^{\text{img}} \tag{1}$$

This loss $\mathcal{L}_t$ will be used to update Faster-RCNN's parameters $\theta_t \to \theta_{t+1}$ for the next input $x_{t+1}$ (see Sec. 3.3). In this section, we will outline each component: pseudo-labelling, image-level recognition, and instance-level recognition.

**Pseudo-labelling** The target image $x_t$ is initially given to the operator and the operator must provide a weak label $z_t$. Then, HL-TTA makes a prediction $\hat{y}_t = \{\hat{\mathbf{b}}_i, \hat{c}_i, \hat{p}_i\}_{i=1}^N = f(x_t\,; \theta_t)$, where $\hat{\mathbf{b}}_i \in \mathbb{R}^4$ is the predicted bounding box, $\hat{c}_i \in \mathcal{C}$ is the predicted object category, $\hat{p}_i \in \mathbb{R}$ is the probability that $\hat{\mathbf{b}}_i$ belongs to $\hat{c}_i$, and $N$ is the total number of predicted bounding boxes. Note that $\hat{y}_t$ is obtained after excluding overlapping boxes by non-maximum suppression for each object category.

However, the prediction $\hat{y}_t$ may contain mistakes, i.e., bounding boxes with incorrect object categories. If we use $\hat{y}_t$ as the groundtruth to adapt $\theta_t$, the errors will accumulate over time, leading to a decrease in the detector's performance. To minimise these errors, we will create a pseudo-label $y_t^{\text{psd}}$ by keeping bounding boxes of $\hat{y}_t$ such that their object categories are present in the weak label $z_t$ and their predicted probability is greater than 0.8

$$y_t^{\text{psd}} = \{\hat{\mathbf{b}}_i, \hat{c}_i \mid \hat{\mathbf{b}}_i, \hat{c}_i, \hat{p}_i \in \hat{y}_t \text{ and } \hat{p}_i \geq 0.8 \text{ and } \hat{c}_i \in z_t\} \tag{2}$$

These pseudo-label $y_t^{\text{psd}}$ and weak label $z_t$ will be respectively used as groundtruth in the instance-level recognition and image-level recognition.

**Image-level recognition** This component aggregates the outputs of RPN and ROI head to obtain an image-level prediction, which is used to calculate the image-level loss $\mathcal{L}_t^{\text{img}}$. This aggregation

operation is developed based on the idea of weakly-supervised object detection (Bilen & Vedaldi, 2016; Xu et al., 2022). Recall that $L$ is the total number of object categories, we denote $K$ as the total number of proposals, the output of RPN as $\mathbf{o} \in \mathbb{R}^K$, and the output of ROI Head as $\mathbf{C} \in \mathbb{R}^{K \times L}$.

Firstly, we create a matrix $\mathbf{O}$ that has a same size as $\mathbf{C}$

$$[\mathbf{O}]_{k,l'} = \begin{cases} [\mathbf{o}]_k & \text{if} \quad l' = \arg\max_l [\mathbf{C}]_{k,1:L} \\ 0 & \text{otherwise} \end{cases} \tag{3}$$

where, $[\cdot]_{k,l}$ denotes the element in the row $k^{\text{th}}$ and column $l^{\text{th}}$ of a matrix, and $[\cdot]_k$ denotes the $k^{\text{th}}$ element of a vector.

Next, the softmax is applied on $\mathbf{C}$ and $\mathbf{O}$

$$[\sigma(\mathbf{C})]_{k,l} = \frac{e^{[\mathbf{C}]_{k,l}}}{\sum_{l=1}^{L} e^{[\mathbf{C}]_{k,l}}}, \quad [\sigma(\mathbf{O})]_{k,l} = \frac{e^{[\mathbf{O}]_{k,l}}}{\sum_{k=1}^{K} e^{[\mathbf{O}]_{k,l}}} \tag{4}$$

Then, the image-level prediction $\hat{z}_t \in \mathbb{R}^L$ is calculated

$$[\hat{z}_t]_l = \sum_{k=1}^{K} \left[ \sigma(\mathbf{C}) \odot \sigma(\bar{\mathbf{O}}) \right]_{k,l} \tag{5}$$

Finally, the image-level loss can be obtained via the standard cross-entropy function

$$\mathcal{L}_t^{\text{img}} = \texttt{cross\_entropy\_loss}\left(\hat{z}_t, \texttt{multi\_hot}(z_t)\right) \tag{6}$$

where, $\texttt{multi\_hot}(\cdot)$ is a function to convert $z_t$ into a multi-hot vector of size $L$.

**Instance-level recognition** This component will employ the pseudo-label $y_t^{\text{psd}}$ from Eq. (2) as the groundtruth. The instance-level loss is formulated as follows

$$\mathcal{L}_t^{\text{ins}} = \mathcal{L}_{\text{cls}}^{\text{rpn}}(x_t, y_t^{\text{psd}}) + \mathcal{L}_{\text{cls}}^{\text{roi}}(x_t, y_t^{\text{psd}}) \tag{7}$$

where, $\mathcal{L}_{\text{cls}}^{\text{rpn}}$ and $\mathcal{L}_{\text{cls}}^{\text{roi}}$ are the classification losses of RPN and ROI Head proposed in standard Faster-RCNN (Ren et al., 2015). Here, the instance-level loss $\mathcal{L}_t^{\text{ins}}$ ignores the bounding-box regression task since the pseudo-label $y_t^{\text{psd}}$ in Eq. (2) indicates the confidence score of predicted bounding boxes. Thus, the parameters $\theta_t$ are adapted to enhance the classification performance of the detector.

### 3.3 Updating batch normalisation

Given the final loss from Eq. (1), we need to adapt $\theta_t \to \theta_{t+1}$ for the next input $x_{t+1}$. We choose to update all BN layers in $\theta$ as this has been shown to be highly effective in recent studies (Mirza et al., 2022; Wang et al., 2021; Schneider et al., 2020). The rationale of updating BN layers is to reduce the covariate shift between the source and target distributions (Schneider et al., 2020). If the target distribution is different from source distributions, BN's parameters estimated from the source distribution are no longer normalising the target data as expected. Therefore, it is necessary to update BN layers with the new target distribution.

Specifically, for an arbitrary BN layer of $\theta_t$, let $\mu_t$ and $\sigma_t$ be its running mean and running variance, and also let $\gamma_t$ and $\beta_t$ be its transformation parameters. We also denote $m_t$ as its momentum ($m_0$ is set to 0.1 by default). As shown in (Mirza et al., 2022), if the momentum is gradually decayed, it will stabilise the convergence of the domain adaptation. Therefore, we initially decay the momentum

$$m_t = m_{t-1}.\omega + \delta \tag{8}$$

where, $\omega \in (0, 1)$ is a predefined decay parameter and $\delta$ defines the lower bound of momentum.

Subsequently, BN's parameters will be updated as follows

$$\mu_{t+1} = (1 - m_t).\mu_t + m.\hat{\mu}_t, \qquad \sigma_{t+1} = (1 - m_t).\sigma_t + m.\hat{\sigma}_t, \tag{9}$$

$$\gamma_{t+1} = \gamma_t + \lambda \frac{\partial \mathcal{L}_t}{\partial \gamma_t}, \qquad \beta_{t+1} = \beta_t + \lambda \frac{\partial \mathcal{L}_t}{\partial \beta_t}, \tag{10}$$

where, $\lambda$ is the step size of gradient update and $\hat{\mu}$ and $\hat{\sigma}$ are mean and variance of the current input data. Note that the previous work (Mirza et al., 2022) only updates $\mu_t$ and $\sigma_t$ while our method can also update transformation parameters $\gamma_t$ and $\beta_t$, thanks to weak labels provided by the operator.

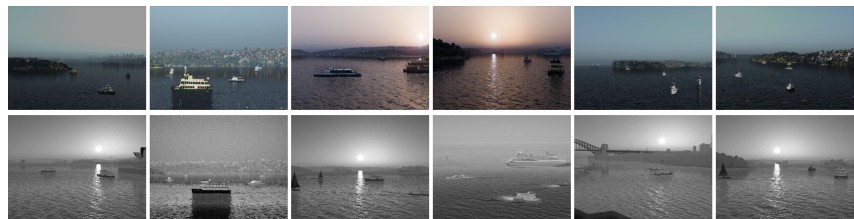

Figure 3: Sample images of our MSA-SYNTH dataset, where top row is RGB images and bottom row is Infrared images.

## 4 EXPERIMENT

### 4.1 SETUP

The following benchmarks are used in the experiment.

- **KITTI → KITTI-Fog**: KITTI (Geiger et al., 2012) is a widely-used dataset in autonomous driving. KITTI is used to pre-train the model, which are then adapted to the target KITTI-Fog with the most severe fog level 30m visibility (Halder et al., 2019). Object categories taken into account are "Car", "Pedestrian", and "Cyclist". A total of 7481 images are randomly divided into 3740 for training and 3741 for testing.
- **Cityscapes → KITTI**: Cityscapes (Cordts et al., 2016) is another popular dataset in self-driving cars. We pre-train the object detector in Cityscape, then adapt it to KITTI. Three object categories in Cityscapes are used: "Car", "Pedestrian", and "Rider". Similarly, three object categories in KITTI are used: "Car", "Pedestrian", and "Cyclist". A total of 3475 Cityscapes images from its training and validation sets are randomly divided into 1737 for training and 1738 for testing. Similarly, a total of 7481 KITTI images are randomly divided into 3740 for training and 3741 for testing.
- **MSA-SYNTH RGB → IR** [1]: We use Unreal Engine and Infinite Studio to generate a maritime dataset. Three boat/vessel categories "Fishing", "Sailing", and "Passenger" are simulated. We collect 8147 RGB images and 8147 Infrared (IR) images, which are then divided into 4243 RGB images for training, 3904 RGB images for testing, 4243 IR images for training, and 3904 IR images for testing; see Fig. 3 for sample images. The model will be pre-trained in the source domain RGB, then adapted to the target domain IR.

We consider following baselines for comparison

- **Source**: The source pre-trained model is tested on the target data without any adaptation.
- **BN stats**: BN stats (Schneider et al., 2020) adapts the source pre-trained model by updating the statistics of batch normalisation (BN) layers.
- **DUA**: DUA (Mirza et al., 2022) introduces a decay factor to update the momentum parameters of the BN layers of the source pre-trained model.
- **Oracle**: The source pre-trained model is fine-tuned in 120k iterations on the target training set with full supervision.

To reduce labour cost, our HL-TTA uses 100 target testing images for adaptation in all benchmarks, unless otherwise stated. For baselines, all target testing images are used for adaptation since they are fully autonomous adaptation techniques. To simulate online adaptation, each target testing image is given to each method one at a time to perform domain adaptation. To measure detection performance, we present the average precision with a threshold of 50% (AP50) for each object category and the mean average precision (mAP) across all object categories.

### 4.2 IMPLEMENTATION

We employ Detectron2 (Wu et al., 2019) for implementation. Faster-RCNN with backbone ResNet-50 is pre-trained on the source dataset with a batch size of 2. Learning rate is initially set to 0.001 for 30k iterations, then reduced to 0.0001 for another 90k iterations. For HL-TTA, unless stated

---

[1] The dataset will be released upon acceptance

otherwise we set $\omega = 0.99$ for KITTI → KITTI-Fog and Cityscapes → KITTI, and $\omega = 0.94$ for MSA-SYNTH RGB → IR. For remaining parameters, unless stated otherwise we set learning rate $\lambda = 0.0001$, $\delta = 0.005$, and $\alpha = 0.1$ for all benchmarks.

## 4.3 RESULTS

**Benefits of human guidance in TTA** As shown in Table 2, domain adaptation significantly enhances object detection performance. Specifically, the fully autonomous TTA methods BN Stats and DUA outperforms Source by 11%-14% mAP in all benchmarks. When human guidance is incorporated into the domain adaptation, HL-TTA increases mAP by 4% mAP in MSA-SYNTH RGB → IR and 1% in KITTI → KITTI-Fog and Cityscapes → KITTI, compared to BN Stats and DUA. The improvement is even more significant for certain object categories. For instance, HL-TTA improves "Car" in KITTI → KITTI-Fog, "Cyclist" in Cityscapes → KITTI, and "Fishing" in MSA-SYNTH RGB → IR by about 3.3%, 2%, and 10% respectively, compared to BN Stats and DUA.

**Effects of noisy weak labels** This experiment examines the possibility of humans providing incorrect weak labels. To simulate this, each element in the groundtruth multi-hot vector has a probability of being switched to an incorrect value. This probability is referred to as the noise ratio. An example of this simulation is shown Fig 4. The results are shown in Fig. 5. In general, HL-TTA can be seen to be sensitive to noisy labels.

At a noise level of 50%, the performance of HL-TTA decreases by about 1% in the "Car" and "Cyclist" categories in the KITTI → KITTI-Fog benchmark. A similar 1% drop is observed in the "Pedestrian" and "Cyclist" categories in the Cityscapes → KITTI benchmark. However, the most significant decrease is seen in the MSA-SYNTH RGB → IR benchmark, where the accuracy of the "Fishing" category decreases by 6% and that of the "Sailing" category drops by 2.5%.

When the noise ratio is increased to 99%, the performance of HL-TTA in KITTI → KITTI-Fog reduces by 6% and 3% in the "Car" and "Cyclist" categories respectively. Similarly, there is a significant decrease of 6% and 5% in the "Car" and "Pedestrian" categories in Cityscapes → KITTI. The most dramatic decline is observed in MSA-SYNTH RGB → IR, where the performance of HL-TTA decreased by more than 8% and nearly 5% in the "Fishing" and "Sailing" categories.

|  | Car | Pedestrian | Cyclist | mAP |
|---|---|---|---|---|
| Source | 23.4 | 26.7 | 12.4 | 20.9 |
| BN Stats | 41.3 | 41.4 | 20.8 | 34.5 |
| DUA | 41.3 | 41.8 | 21.3 | 34.8 |
| HL-TTA | 44.6 | 41.9 | 23.1 | 36.5 |
| Oracle | 85.5 | 65.7 | 68.3 | 73.2 |

(a) KITTI → KITTI-Fog

|  | Car | Pedestrian | Cyclist | mAP |
|---|---|---|---|---|
| Source | 66.9 | 46.4 | 9.0 | 40.8 |
| BN Stats | 68.1 | 50.1 | 12.3 | 43.5 |
| DUA | 68.1 | 50.3 | 12.7 | 43.7 |
| HL-TTA | 68.1 | 51.5 | 14.3 | 44.6 |
| Oracle | 90.4 | 70.7 | 77.2 | 79.4 |

(b) Cityscapes → KITTI

|  | Fishing | Sailing | Passenger | mAP |
|---|---|---|---|---|
| Source | 42.4 | 15.6 | 33.5 | 30.5 |
| BN Stats | 43.8 | 14.8 | 37.0 | 31.8 |
| DUA | 44.6 | 15.1 | 37.0 | 32.2 |
| HL-TTA | 54.7 | 21.1 | 36.2 | 37.4 |
| Oracle | 69.9 | 39.2 | 72.8 | 60.6 |

(c) MSA-SYNTH RGB → IR

Table 2: Comparing AP50 within each object categories and mAP across all categories between HL-TTA and other baselines (larger is better).

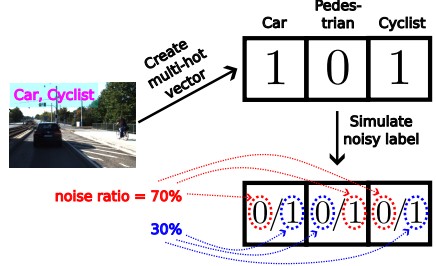

Figure 4: Illustration of how noisy weak labels are simulated. Given a weak label {Car, Cyclist}, a corresponding multi-hot vector is created. If the noise ratio is 70%, the value 1 in "Car" element will have the 70% probability of being switched to 0, while having the 30% probability of remaining 1. A similar operation is applied to elements "Pedestrian" and "Cyclist".

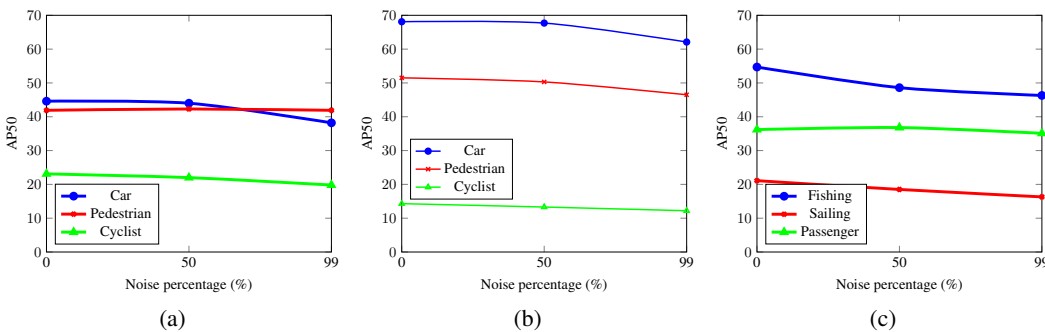

Figure 5: Effects of noisy weak labels to HL-TTA are shown on benchmarks **(a)** KITTI → KITTI Fog, **(b)** Cityscapes → KITTI, and **(c)** MSA-SYNTH RGB → IR.

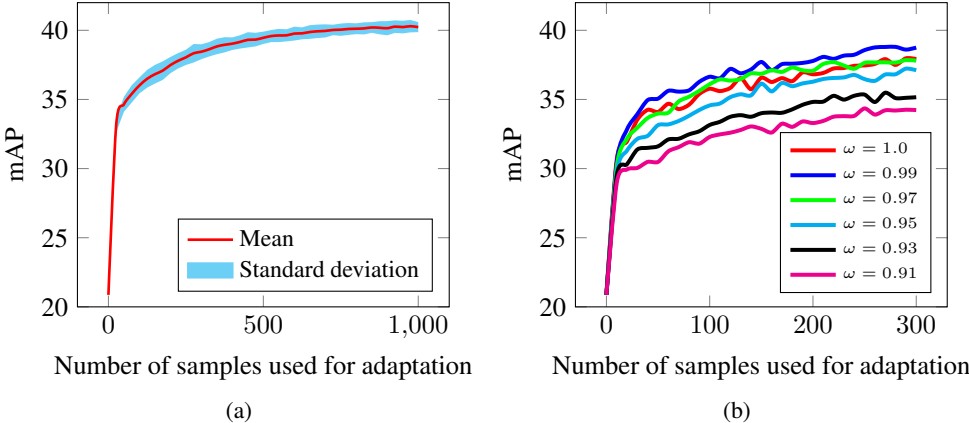

Figure 6: **(a)** HL-TTA results on KITTI → KITTI-Fog for 30 independent runs. For each run, the order of KITTI-Fog testing samples is randomly shuffled. **(b)** HL-TTA results on KITTI → KITTI-Fog for different decay factors.

**Effects of sample orders** This experiment investigates the effects of sample orders. The result on KITTI → KITTI-Fog in shown in Fig. 6a, where we conduct 30 independent runs and calculate the mean and standard deviation of mAP. For each run, the order of samples is randomly shuffled. In the first few samples used for adaptation, the standard deviation is large. For example, with 100 samples, we obtain mAP of $36.0 \pm 0.6$. When the adaptation samples increase, the mAP continues to improve as well as the standard deviation decreases. For instance, the mAP achieves $39.8 \pm 0.5$ at 600 samples and $40.2 \pm 0.4$ at 1000 samples. However, it can be seen that the mAP saturates at 40.0 after 700 samples.

**Effects of decay factors** We investigate the effect of the decay factor $\omega$ on the performance of HL-TTA on the KITTI → KITTI-Fog benchmark. Fig. 6b shows the results when different decay factors $\omega$ from 1.0 (no decay) to 0.91 are applied. We observe that when $\omega$ is set to 0.99 or 0.97, the mAP is better than that of $\omega = 1.0$, indicating that decaying the momentum accelerates the convergence of domain adaptation. However, if the momentum decays too quickly (i.e. $\omega < 0.97$), the detection accuracy decreases. This suggests that tuning the decay factor $\omega$ is essential to achieve the optimal domain adaptation performance.

## 5 CONCLUSION

This paper presents a method involving a human operator in TTA. The algorithm only requires the operator to provide weak labels for images, which are then used to guide the adaptation process. The experiments show that the proposed method outperforms existing autonomous test-time adaptation solutions, demonstrating great potential of human guidance for TTA.

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
