# OpenReview forum: "Human-in-the-Loop Test-Time Domain Adaptation for Object Detection"
_ICLR.cc/2024/Conference — ICLR 2024 Conference Withdrawn Submission_

### Official Review · Reviewer_8iDr · 2023-10-23

**Soundness:** 2 fair
**Presentation:** 3 good
**Contribution:** 2 fair
**Rating:** 3
**Confidence:** 3

**Summary:**

In this paper, the authors propose a simple mechanism to perform test-time domain adaptation for object detection. To be specific, the authors suggest taking weak-labels from a human operator and filtering the predictions of a pretrained detector with those labels. Then, an image-level weakly-supervised OD and instance-level fully-supervised OD (with the filtered pseudo-labels) are trained. Results are reported on KITTI, Cityscapes and a new dataset introduced by the authors.

**Strengths:**

+ Test-time adaptation is an important problem in OD.

**Weaknesses:**

1. In the paper, a pre-trained detector's predictions are simply filtered with the weak labels provided by the operator. This is the only novelty in the paper and I find it rather weak, considering also the minor performance gaps compared to the alternative architectures.

2. It is not clear why a new dataset is curated for this problem.

3. It would have been nicer to include results using the commonly-used COCO dataset.

4. The approach is rather expensive, considering that there are three detectors (one frozen, one for WSOD, and one for FSOD). I would expect discussion on this complexity and also potential reduction/experiments on combining at least WSOD and FSOD.


Minor comments:
- "full automated" => "fully automated".

**Questions:**

Please see Weaknesses.

---

### Official Review · Reviewer_28Gn · 2023-10-28

**Soundness:** 2 fair
**Presentation:** 2 fair
**Contribution:** 1 poor
**Rating:** 3
**Confidence:** 4

**Summary:**

The authors explore test-time domain adaptation for object detection, traditionally where object detection models adapt to the settings seen in a given use case or set of test data without labels. They explore introducing human input (labels) into the process on the test data to improve the accuracy of the adaptation. They require only weak labels, which reduces the human time required to assist inadaptation. They show that having some human-generated labeling signal at test time improved performance, which is a very intuitive result as the model is being provided a much more useful signal than it gets without human labels. Performance is still significantly lagging behind the “oracle” performance, which assumes a large amount of strongly-labeled test-domain data.

**Strengths:**

The paper clearly motivates their target use case by pointing out that many test-time adaptation scenarios do not need to be fully automated, and that in order to achieve accuracies that are good enough to be useful and usable at test time some human supervision may be required. They show that labeled signal, even when it is only a weak label, improves performance slightly for several domain adaptation benchmarks.

**Weaknesses:**

The distinction between weakly-supervised domain adaptation and human-in-the-loop test-time adaptation is very slight, and possibly the only methodological difference as defined by the authors is that weakly-supervised systems assume that the human-in-the-loop is offline, which I’m not sure is a universal assumption in this setting. Furthermore, the only change to these methods to make them “online” and identical to the proposed method is to assume iteration or additional human labels are allowed to be requested. There is also a large body of work on active learning for domain adaptation that is not covered in the literature and is not clearly a different setting than what the authors propose, particularly when object detectors are being trained (for example https://zslpublications.onlinelibrary.wiley.com/doi/full/10.1002/rse2.200, ). Given this literature, the novelty of the work is a bit more unclear.

In particular, they do not compare to any alternate active learning or human-in-the-loop benchmarks, which makes in unclear whether their proposed method’s performance increase is fundamentally rooted in the additional labels or is actually tied to something about the method itself. They also do not compare to alternative uses of that human time, which would significantly strengthen the claim that their method is superior in these settings where some human time budget is available.

**Questions:**

What would a fair comparison benchmark be for your method that uses perhaps a simpler adaptation architecture but assumes the exact same human input. For example, what is the performance of the source model when the weak class labels are used to filter the detections?

There is still a significant gap between the proposed method and the oracle. What is the reason for this gap when human labels are being incorporated? Just the number of training examples with human input? The weak vs strong input? Instead of requiring weak human labels for test frames, how might that human budget be used differently? For instance, providing a strong label for every k frames by correcting the source model detections, where k is set such that the human time is the same? Is it truly better to get a lot of weak labels for adaptations vs a smaller set of strong ones?

The improvements from the human-in-the-loop adaptation seem small in most cases. How might human time be better accounted for in the metrics or provided as context? What is the value of that human time budget spent on adaptation with class-level supervision for a set of samples vs just spent doing efficient quality control/verification/correction or choosing an operating threshold per-category based on a specific test case?

How does performance in the “online” setting they propose differ from performance in a 1-shot weakly-supervised adaptation setting, where you would collect k samples and label them (weakly or strongly) and then tune the model on those k samples? What makes this system truly “online” for these benchmarks vs the prior work?

---

### Official Review · Reviewer_Dn7S · 2023-11-01

**Soundness:** 3 good
**Presentation:** 3 good
**Contribution:** 2 fair
**Rating:** 5
**Confidence:** 5

**Summary:**

In this paper, the authors proposed a Human-in-the-loop test-time domain adaptation (HL-TTA) that is able to take advantage of weary labels from human to update the object detector model during the testing phase.
Given a few samples with weak labels (i.e. only the object categories without bounding boxes), HL-TTA framework firstly produces pseudo-labels for these testing samples. Then, image-level and instance-level recognition losses are adopted to update the batch normalization layers of the network.
Experiments on several datasets have shown the advantages of HL-TTA.

**Strengths:**

- The paper is well-motivated.
- The idea of human-in-the-loop and online updating during testing time is interesting and potential.
- Experimental results show the advantages of the proposed approach in several aspects such as noise level, sample orders, decay parameters.

**Weaknesses:**

1. While I acknowledge the interesting motivation of HL-TTA, the novelty is limited and incremental.
Particularly, the idea of human-in-the-loop domain adaptation has been presented in literature. Moreover, all the steps, i.e. pseudo-labeling; image-level recognition and instance-level recognition losses; batch norm updating are also adopted from previous works.

2. In Pseudo-labeling step, the authors proposed to keep bounding boxes of objects with the probability of 0.8. What happens if the detector cannot give any confident prediction (i.e. probability is less than 0.5) in the testing environment? In that case, y_t^{psd} is empty and no learning can be employed.

3. How can HL-TTA prevent catastrophic forgetting?

**Questions:**

Please address the concerns in the Weakness Sections.

---

### Official Review · Reviewer_ANjN · 2023-11-10

**Soundness:** 3 good
**Presentation:** 3 good
**Contribution:** 2 fair
**Rating:** 5
**Confidence:** 5

**Summary:**

The paper tackles continuous adaptation of object detectors to to unforeseen scenarios. The proposed method unlike existing methods that focus fully on adaptation from unlabeled data, involves human operators and performs a weakly labeled domain adaptation. The specific target of this method is scenarios like surveillance, self-driving, and demonstrates the efficacy of human-in-the-loop signal test-time domain adaptation.

**Strengths:**

+ The paper is meticulously written, well structured, and tackles the really interesting and challenging problem of continuous domain adaptive detection. The problem motivation is clearly explained in the introduction section and connects well to the proposed approach.
+ The paper has good flow making it easier to follow. The paper points out a missing element in the existing literature, and provides a different way of formulating the adaptive detection problem, with human signal in the loop.
+ The idea behind the proposed method to utilize human supervision is not directly used in the literature especially not for object detection, hence it is definitely an idea worth exploring. More about the methodology in the next section.
+ The paper provides numerous experiments under different scenarios to test the robustness of the adaptation strategy under incorrect human guidance and adaptation conditions. The experiments supports the motivation that weak label from human guidance improves the performance on unseen domain.

**Weaknesses:**

+ Though the problem is quite interesting and challenging, the proposed method is limited in its effectiveness to fully utilize the proposed framework of human in the loop TTA. The human supervision is directly used as filter, where as there is a lot of research in the domain of active learning and weakly supervised learning which have explored incorporating such supervision in much more interesting way. Weak-label to improve pseudo-label quality has been tested multiple times over in the literature.
+ There are not enough datasets and scenarios considered in the experimental evaluation. Specifically considering the motivation on real-world conditions, the domain does not remain constant and is subjected to change as well. Hence, a mixing of domain during adaptation would reflect a more realistic and challenging scenario
+ There are not enough test-time methods compared against to properly benchmark the performance. There have been many work in the literature that address a similar problem [1] [2] [3] [4].

[1] Mirza, Muhammad Jehanzeb, et al. "ActMAD: Activation Matching to Align Distributions for Test-Time-Training." Proceedings of the IEEE/CVF Conference on Computer Vision and Pattern Recognition. 2023.
[2] Yuan, Jiakang, et al. "Bi3D: Bi-domain Active Learning for Cross-domain 3D Object Detection." Proceedings of the IEEE/CVF Conference on Computer Vision and Pattern Recognition. 2023.
[3] VS, Vibashan, Poojan Oza, and Vishal M. Patel. "Towards online domain adaptive object detection." Proceedings of the IEEE/CVF Winter Conference on Applications of Computer Vision. 2023.
[4] Suzuki, Satoshi, et al. "OnDA-DETR: Online Domain Adaptation for Detection Transformers with Self-Training Framework." 2023 IEEE International Conference on Image Processing (ICIP). IEEE, 2023.

**Questions:**

**Q1.** Since the motivation is on for a system that is deployed in real-world, what is the latency increase with using proposed method for improving generalization on unseen domains?

**Q2.** what is the performance variations when only part of the samples utilize human annotations? The experiment seem to be only considering noisy supervision and not the absence of it.

**Q3.** In section 4.3 experiment related to weak-label noise why some categories have more performance drop compared to others?